# Platelet-Derived S1P and Its Relevance for the Communication with Immune Cells in Multiple Human Diseases

**DOI:** 10.3390/ijms231810278

**Published:** 2022-09-07

**Authors:** Céline Tolksdorf, Eileen Moritz, Robert Wolf, Ulrike Meyer, Sascha Marx, Sandra Bien-Möller, Ulrike Garscha, Gabriele Jedlitschky, Bernhard H. Rauch

**Affiliations:** 1Division of Pharmacology and Toxicology, School of Medicine and Health Sciences, Carl von Ossietzky University Oldenburg, 26129 Oldenburg, Germany; 2Department of General Pharmacology, University Medicine Greifswald, 17489 Greifswald, Germany; 3Department of Neurosurgery, University Medicine Greifswald, 17489 Greifswald, Germany; 4Institute of Pharmacy, University of Greifswald, 17489 Greifswald, Germany

**Keywords:** sphingosine-1-phosphate, platelets, immune cells, S1P, S1P receptors

## Abstract

Sphingosine-1-phosphate (S1P) is a versatile signaling lipid involved in the regulation of numerous cellular processes. S1P regulates cellular proliferation, migration, and apoptosis as well as the function of immune cells. S1P is generated from sphingosine (Sph), which derives from the ceramide metabolism. In particular, high concentrations of S1P are present in the blood. This originates mainly from erythrocytes, endothelial cells (ECs), and platelets. While erythrocytes function as a storage pool for circulating S1P, platelets can rapidly generate S1P de novo, store it in large quantities, and release it when the platelet is activated. Platelets can thus provide S1P in a short time when needed or in the case of an injury with subsequent platelet activation and thereby regulate local cellular responses. In addition, platelet-dependently generated and released S1P may also influence long-term immune cell functions in various disease processes, such as inflammation-driven vascular diseases. In this review, the metabolism and release of platelet S1P are presented, and the autocrine versus paracrine functions of platelet-derived S1P and its relevance in various disease processes are discussed. New pharmacological approaches that target the auto- or paracrine effects of S1P may be therapeutically helpful in the future for pathological processes involving S1P.

## 1. Introduction

Sphingolipids belong to the class of polar lipid compounds. They are important structural components of the cell membrane and exert various signaling functions, either intracellularly as second messengers or via membrane-bound receptors. The family of sphingolipids consists of three main types: the ceramides, sphingomyelins (SMs), and glycosphingolipids. Glycosphingolipids are further divided into the cerebrosides and gangliosides. These types differ in the nature of their respective chemical moieties [1]. In contrast to glycerol-based phosphoglycerides, sphingolipids are derived from the unsaturated amino alcohol sphingosine (Sph) [2]. The phosphoric acid ester of Sph is sphingosine-1-phosphate (S1P), which is found in high concentrations in the lymph fluid and in the blood. The main producers of circulating S1P are erythrocytes, endothelial cells (ECs), immune cells, and platelets. Within organ tissues, S1P is rapidly degraded by the action of S1P lyase (SPL). As a result, there is a steep gradient between high S1P concentrations within the vessels and low S1P concentrations in the tissue [3].

S1P exerts its multiple functions via a family of G-protein-coupled receptors (GPCRs), the S1P receptors 1-5 (S1PR1-5). S1PRs exhibit a tissue-specific expression and possess partly synergistic and antagonistic functions. Almost all body cells appear to express S1PRs. Recent studies indicate that S1P signaling regulates many more cell types and processes than previously appreciated [4]. Typical receptor-mediated functions of S1P relate to cell proliferation, migration, and apoptosis [4]. In particular, S1PRs are highly expressed in immune cells and regulate functions in inflammatory diseases and wound healing [5,6]. S1P exerts key roles during immune cell responses, e.g., on B cell migration and their interaction with T cells. It also affects the positioning of immune cells, such as natural killer cells, within lymph nodes and regulates their response to interferon-γ [7,8]. In addition, it has recently been shown that monocyte-derived S1P in the lymph nodes regulates immune responses [9].

The S1P gradient is essential for maintaining physiological tissue homeostasis. It guides cells from the low-S1P environment of tissues into the high-S1P environment of circulatory fluids [6]. Disturbances in the S1P gradient lead to altered migration behavior of immune cells and other cell types. Local changes in the S1P gradient, for example, triggered by a local injury with a subsequent inflammatory reaction, regulate the activity and migration behavior of immune cells and the cells of the vessel wall. From a pathophysiological point of view, the local change in the S1P gradient thus supports the reaction of the immune system and strengthens the local defense and healing processes. The importance of platelets, platelet-derived S1P, and the S1PRs involved in inflammatory responses are addressed in the present review.

## 2. S1P-Biosynthesis, Release, and Functions

S1P is formed by the phosphorylation of Sph. Several sources of Sph have been described. It is either derived from plasma and incorporated into vascular circulating blood cells or is generated at the outer leaflet of the plasma membrane, initiated by cell surface SM degradation [10]. In addition, Sph formation results from the enzymatic activity of ceramidase (CDase) from ceramides. Ceramides, in turn, can be generated from SM, glycosphingolipids, and de novo from serine and palmitoyl-coenzyme A (palmitoyl-CoA) [11,12]. Ceramides and S1P have different, if not opposing, cell-regulating functions. While ceramides can cause cell apoptosis, S1P is more likely to have an anti-apoptotic function [13]. The S1P present in the blood or tissue can either be irreversibly degraded into phosphoethanolamine and hexadecenal by the SPL or can be dephosphorylated by S1P phosphatases (SPPs) [14]. The generation of S1P from Sph is achieved by two different sphingosine kinases (SphK1 and SphK2). The expression of the SphKs varies from tissue to tissue. While SphK1 is most abundant in the lung, spleen, and liver, SphK2 is expressed predominantly in the liver and in the heart [12]. SphKs modulate the balance between S1P, Sph, and ceramides. As a result of this enzymatically balanced homeostasis, the physiological levels of Sph and ceramide remain stable [15].

The release of intracellularly generated S1P to the extracellular space and into the blood has been investigated by several studies. While sphingolipid transporter 2/spinster homolog 2 (Spns2) has been reported to facilitate the release of S1P from vascular ECs [16], other investigators found that Spns2 is required for the secretion of lymph but not plasma S1P [17]. Another transporter essential for S1P export from red blood cells and platelets is the major facilitator superfamily transporter 2b (Mfsd2b) [18]. In mast cells and certain tumor types such as breast cancer, S1P can be released by members of the adenosine triphosphate (ATP)-binding cassette (ABC) transporter family [19,20]. In platelets, results from our group suggest that ABCC4/MRP4 (multidrug resistance protein 4) is involved in the secretion of S1P [21]. This may particularly become relevant under conditions of pathophysiological platelet activation, e.g., during injury, to elevate local S1P availability to reinforce local inflammatory responses.

About 60% of total S1P circulation in the blood is stored in the lipoprotein fraction of plasma and serum [22]. High-density lipoprotein (HDL)-associated S1P is bound specifically to apolipoprotein M (apoM) [23]. The remaining amount of S1P (30–40%) is bound and transported via albumin (alb), low-density lipoproteins (LDLs), and very low-density lipoproteins (VLDLs) [24]. 

The S1P levels in plasma range between 0.1 and 1.2 µM, while S1P serum concentrations range between 0.5 and 1.2 µmol [25,26,27]. On average, men appear to exhibit higher S1P serum concentrations than women [27]. Depending on its cellular localization and the roles of the different S1PRs, the signaling lipid can exert various functions [28]. For example, the S1P delivered by apoM to S1PR1 on ECs has vascular protective functions [23]. Further functions of S1PRs will be discussed later. A scheme of the synthesis pathway in general, the release to the extracellular space, and the binding partners of circulating S1P is shown in Figure 1.

### 2.1. S1P Receptors and Functions

As an autocrine or paracrine mediator, S1P unfolds its various effects by binding to ligand-specific receptors. S1P binds to five GPCRs, S1PR1–5 [29]. This causes the activation or inhibition of signal cascades in different cell types with partly overlapping or divergent functions. For example, S1PR1 mediates mainly physiological functions, while S1PR2 is believed to exert a key role in inflammatory processes such as arteriosclerosis [30,31].

GPCRs are linked to three different subunits: Gα, Gβ, and Gγ. Each subunit activates a different effector system [32]. S1P mediates its effect after binding to the S1PR via different α units (α_i_, α_q_, and α_12/13_) [33]. The activation of the Gα_i_ subunit activates the adenylylcyclase, leading to a reduction in cyclic adenosine monophosphate (cAMP). The αi subunit of S1PR mainly influences migration, proliferation, vessel formation [34,35], and inflammation [36]. In comparison, the Gq-mediated activation of phospholipase C (PLC) increases the intracellular Ca^2+^ (calcium) concentration. Further downstream signaling of S1P is mediated via the activation of the extracellular signal-regulated kinase 1 and 2 (ERK1/2). Other effects are triggered through the activation of the α_12/13_ subunit. This pathway involves downstream signaling through Ras homolog gene family member A (RhoA) and RAS-related C3 botulinum toxin substrate 1 (Rac1), which reduces cell migration and proliferation but promotes inflammatory processes [34,37].

Studies indicate that S1PRs are involved in receptor cross-activation. This is particularly relevant for different cellular functions and the crosstalk of signaling pathways. One possible mechanism is the dimerization of S1PRs [38]. Van Brocklyn et al. suggested the dimerization of S1PRs1–3 [38]. This was confirmed by the work of Zaslavsky et al. in which S1PRs1–3 formed heterodimers with lysophosphatidic acid (LPA) receptors 1-3 and GPR4 [39]. S1PR activation can also activate other receptors, such as the chemokine receptor 4 (CXCR4) in endothelial progenitor cells (EPCs). Walter et al. confirmed an S1P-induced activation of CXCR4 signaling through S1PR3. In this regard, the phosphorylation of the CXCR4 receptor is stimulated by S1P and, as a result, enhances the angiogenic and neovascularization capacity of EPCs [40].

### 2.2. Platelet-Derived S1P

In the event of a blood vessel injury, platelets have the function of covering the wound and stopping blood loss. Various mechanisms are set in motion for this purpose. First, platelets are activated by von Willebrand factor (VWF), and they subsequently adhere to the extracellular matrix. Autocrine and paracrine mediators such as adenosine diphosphate (ADP), thrombin, or epinephrine amplify and sustain the initial platelet response, leading to an activation of integrin glycoprotein IIb/IIIa (αIIbβ3). Circulating platelets are recruited and form a hemostatic plug [41]. Platelet activation is very complex. Consequently, different mechanisms are involved in activation. Among other things, platelet mitochondria may influence platelet activation and modify the platelet response to stimulation [42].

A lipid that is involved in the autocrine and paracrine activation of platelets is S1P. As described above, S1P is formed from Sph. Results from Tani et al. suggest a limited capacity for the de novo synthesis of sphingolipids within platelets, since platelets barely show serine palmitoyltransferase activity [10]. Platelet Sph, and in turn S1P, is derived from circulating SM through different reactions involving sphingomyelinase (SMase) and CDase, either in plasma or on the platelet plasma membrane [10]. In contrast to erythrocytes, which are devoid of SphK, the SphK activity is proportional to the number of platelets [43]. Here, the lipid is formed by the phosphorylation of Sph through SpK2, which has been reported as the quantitatively dominant isoform in platelets [44,45]. In addition, existing S1P can be dephosphorylated by an ectophosphatase, (possibly lipid phosphate phosphohydrolase (LPP)) to Sph, which then can be incorporated into platelets and therefore infiltrated into the cycling pathway of S1P generation in platelets [46].

Since platelets lack the degrading enzyme SPL, platelets store large amounts of the lipid [47] and cause platelets to be the main source of locally elevated S1P levels after activation [48]. S1P is stored either in the platelet plasma membrane or, presumably, within granules. While S1P stored in the plasma membrane is more likely to be the metabolically active pool, the S1P stored in granules seems to function as the main source of secreted S1P after platelet activation and degranulation [49]. It is currently not fully clear whether stimulated platelet S1P release mainly occurs from these intracellular storage pools or also, in part, as a result of elevated SphK activity upon platelet activation, e.g., by the translocation of SphK to the plasma membrane. This has been suggested recently for SphK1 in fibroblasts [50]. In platelets, SphK2 has been reported as the quantitatively relevant isoform, and the question of whether it is translocated to the plasma membrane upon platelet activation has not yet been resolved. Subsequent to its release, platelet-derived S1P binds in an autocrine manner to its receptors located on platelets or in a paracrine manner to vascular ECs and other blood cells [28]. Figure 2 summarizes the current understanding of the different storage pools of S1P in circulating nonactivated platelets, which are often called “resting” platelets—although this is the active energy-consuming state [51]—and the release of S1P from agonist-activated platelets.

On the one hand, in some studies, S1P has been reported to directly affect platelet aggregation, but on the other hand, it is also crucial for the formation of new platelets [52]. However, the precise role of S1P in thrombopoiesis remains controversial. While data from mouse models indicate that mice lacking S1PR1 develop severe thrombocytopenia caused by both the formation of aberrant extravascular proplatelets (PPs) and defective intravascular PP shedding [52], other groups more recently found that S1P may restrict megakaryopoiesis through S1PR1 and may further suppress thrombopoiesis through S1PR2 when aberrantly secreted in the hematopoietic niche [53]. This observation is in agreement with the clinical use of S1P pathway modulators, including fingolimod (FTY720), a multiple sclerosis (MS) therapeutic, which has not been associated with the risk of bleeding or thrombosis [53]. The exact mode of action of fingolimod and specifically its effect on platelet function will be discussed in more detail in Section 3.3.

Platelets are also able to release extracellular vesicles (EVs), which serve as lipid shuttles and mediate inflammatory processes such as lung injury [54,55]. EVs accumulate during platelet storage and become particularly relevant in the development of transfusion-related acute lung injury (TRALI), supposedly due to their high volume of long-chained ceramides and S1P depletion. This was discovered by the work of McVey et al. in which the sphingolipid rheostat was assayed. The EVs in human or mouse plasma-derived platelets stored for 5 days in comparison to plasma stored for 1 day showed an almost complete loss in S1P and induced an endothelial monolayer barrier leak in vitro which, for murine platelet-derived EVs, induced the characteristic symptoms of TRALI in vivo. Therefore, ceramides may participate in EV-mediated lung injury, while S1P has anti-apoptotic effects and exerts barrier-protective properties [54].

The S1P secretion mechanisms of platelets and the resulting autocrine effects have yet to be clarified. There is evidence for an ATP-dependent export of S1P from the cytoplasm across the plasma membrane mediated by MRP4, which is based on transport studies in isolated membranes, fluorescence microscopy using labelled S1P, and the reduced release of S1P from murine MRP4-deficient platelets [21,56]. In resting platelets, the microscopy studies also suggested a partial colocalization of MRP4 and S1P in granular structures, which may represent dense granules [21]. Consequently, in resting platelets, MRP4 is probably located in part in the membrane of granules and is fully incorporated into the plasma membrane upon platelet activation [57]. Decouture et al. suggested a role of MRP4 in the granular S1P storage [58], and Jonnalagadda et al. reported S1P storage in α-granules [49]. An earlier publication already differentiated between an ATP-dependent release stimulation by thrombin and an ATP-independent S1P release via Ca^2+^ influx [59]. Findings also suggested an involvement of protein kinase C (PKC) in S1P release from platelets [47]. At present, it is unclear how PKC, platelet degranulation, and MRP4 are linked together (see scheme in Figure 2).

In 1995, Yatomi et al. first described S1P as a possible platelet agonist. This has been critically discussed by different groups and is still not fully clarified to date. In their study, S1P alone induced platelet aggregation and shape change. Platelet aggregation may have been triggered through a Ca^2+^ increase induced by S1P, which was also shown in this study [60]. In agreement with these findings, Randriamboavonjy et al. showed a concentration-dependent increase in cellular Ca^2+^ levels after the S1P stimulation of washed platelets [61]. However, this assumption was questioned by the work of Nugent et al. [62]. Here, S1P did not show an effect on platelet aggregation, and no synergistic effect on ADP-induced platelet activation was described [62]. Other studies also confirmed platelet agonistic effects after S1P binding to its receptors. Urtz et al. showed platelet activation through S1P/S1PR1 binding in whole blood [45].

Besides its release from platelets, authors have described an activity-modulating effect of S1P via binding to S1PR1, which alters platelet aggregation through protease-activated receptor 4 (PAR-4) or ADP [45,60]. In human platelets, the activation of PAR-1 by thrombin stimulates a potent release of S1P. This stimulated S1P release appears to depend at least partially on thromboxane (TX) synthesis and the consecutive activation of the thromboxane receptor (TP) [63]. In addition, the activation of S1P release can also be provoked by agonists that lead to TX levels high enough to activate TP signaling, such as high collagen concentrations. The key role of TP receptor activation for bulk S1P secretion was confirmed by the use of a classical nonselective inhibitor of cyclooxygenase (COX), i.e., acetylsalicylic acid (ASS) or ibuprofen, which significantly reduce platelet-derived S1P release [63]. Recent studies from Liu et al. also confirmed a synergistic effect of S1P and PAR-1 activation. However, aggregation was affected in a concentration-dependent biphasic manner. While the coincubation of 100 nM S1P and the PAR-1-activating peptide SFLLRN significantly increased peak aggregation, coincubation with 10 µM and 30 µM S1P and PAR-1 peptide reduced platelet aggregation. Using specific agonists and antagonists, findings suggest that S1PR1 activation enhances platelet aggregation, while S1PR4 and 5 may mediate inhibitory effects [64]. Interestingly, other studies also suggest a role for S1PR2 in platelet activation and link it to the functions of the RhoA-Rho kinase pathway [61]. The downregulation of S1PR2 in type 2 diabetes has been reported to be associated with attenuated responses to S1P. However, the pathophysiological meaning of this mechanism is currently unclear [61].

Studies from Onuma et al. suggest an inhibitory function of S1P in collagen-induced platelet aggregation. Here, S1PR4 appears to be predominantly involved, as S1PR1 did not show any difference in activation. S1P alone did not affect platelet aggregation [65]. Table 1 summarizes the effects of S1P on platelets and megakaryocytes (MKs).

Taken together, the function of S1P on platelet aggregation remains controversial (see scheme in Figure 3). However, it seems as if S1P does not function as a classical direct agonist but instead may contribute to controlling platelet responses to a variety of stimuli [28].

### 2.3. S1P—Immune System and Platelets

In addition to its influence on platelet activation, S1P affects a variety of cellular functions, including immune cell migration and cytokine expression. In particular, within the immune system it has been shown that human dendritic cells (DCs) express mRNA for S1PR1, S1PR2, S1PR3, and S1PR4 [67]. DCs are part of the innate immune system and are specialized antigen-presenting cells with the ability to migrate into peripheral tissues and lymph nodes as well as to control the activation of naïve T cells. In maturing DCs, S1P inhibits the secretion of tumor necrosis factor (TNF)-α and interleukin (IL)-12. In contrast to this, it enhances the secretion of IL-10 [67]. While TNF and IL-12 both have pro-inflammatory effects, IL-10 limits immune responses and therefore functions as an anti-inflammatory mediator [68,69,70]. Hence, S1P functions as both a pro- and anti-inflammatory mediator and shows a reduced and increased capacity to generate T helper type 1 (Th1) and T helper type 2 (Th2) cell responses. S1P induces PAR-1 and PAR-4 mRNA and protein expression on primary monocytes and monocytic cells in vitro. In particular, the activation of S1PR1 and S1PR3 seem to play a role in the regulation of PAR expression. The upregulation of PAR-4 is triggered via S1P binding to S1PR3. Moreover, the activation of S1PR3 leads to an increased monocytic migration towards thrombin. These responses were associated with phosphatidylinositol-3-kinase (PI3K) -mediated COX-2 induction and consecutively elevated prostaglandin E2 (PGE2) production [71]. S1PR3 also mediates the release of inflammatory factors and therefore participates in inflammatory processes. In macrophages, S1PR3 production is stimulated by lipopolysaccharides [72]. Cell adhesion molecules expressed by activated ECs are important for the trafficking of circulating immune cells across the EC barrier into the underlying tissue [73]. Further, S1P concentrations of 1 µM stimulate the migration of ECs by S1PR1 and S1PR3 [74]. Other studies showed the prevention of monocytic adhesion to ECs from type 1 diabetic NOD (non-obese diabetic) mice due to decreased intercellular adhesion molecule-1 (ICAM-1) expression in response to the activation of S1PR1 with low S1P concentrations of 0.1 µM [75]. These findings suggest S1PR1 on ECs plays a crucial role in regulating monocyte adhesion and chemotaxis to ECs. Depending on the concentration of S1P, it can function in either an anti-inflammatory or pro-inflammatory manner. Regarding the monocyte-EC interaction, umbilical vein endothelial cells (HUVECs) treated with S1P concentrations of 5 µM mediate the chemotaxis of transformed human mononuclear (THP-1) cells (which are human leukemia monocytic cells) to HUVECs. S1P upregulates ICAM-1, IL-8, and monocyte chemoattractant protein-1 (MCP-1) expression in ECs through S1PR1 [73,76]. ICAM-1 is a cell surface glycoprotein and an adhesion receptor that regulates leukocyte recruitment from the circulation to sites of inflammation [77].

An important function of S1P in the immune system is to control lymphocyte migration and circulation through S1PR1. S1P concentrations of 3–30 nM promote lymphocyte movement from lymph nodes into efferent lymph vessels and then to the blood [78]. In mice, it has been described that the S1P transporter Spns2 regulates the egress of mature T cells and immature B cells from the thymus and bone marrow [79]. S1P protects T cells from apoptosis and enhances their chemotaxis to chemokines and other chemotic factors [78]. These findings remain controversial since other studies demonstrated attenuated chemotaxis in response to chemokines due to the activation of S1PR1 by S1P. Nevertheless, S1P also evoked T cell chemotaxis at 1–100 nM [80].

Polymorphonuclear neutrophils (PMN) are circulating blood leukocytes that provide the first line of defense against infection and effectors of inflammation [81]. PMN react to S1P by a Fcγ receptor (FCγR) -mediated rise in intracellular Ca^2+^ and reactive oxygen species (ROS), including shape change and the adhesion of PMN to immune-complex-coated surfaces [82]. Antibodies regulate immune responses via interacting with FCγR, leading to the activation of innate effector cells or the maturation of DCs [83]. Modulating the response of FCγR to antibody binding may represent another mechanism for S1P to amplify leukocyte responses and recruitment.

As described previously, SMases are a family of enzymes that hydrolyze sphingomyelin to ceramides [84]. The release of acid SMase (A-SMase) by ECs and secretory SMase (S-SMase) by macrophages is triggered by inflammatory stimuli such as IL-1β and interferon-γ. SMase activity as well as CDase and SphK activities lead to S1P production in platelets and therefore increasing S1P levels in platelets during inflammatory processes [28]. Platelets adhere to an inflamed endothelium and enhance leukocyte recruitment and activation. Platelet-secreted S1P regulates leukocyte trafficking and attracts immune cell adhesion to the endothelial layer, thereby elevating local immune cell recruitment and the production of pro-inflammatory mediators [28]. Comparable mechanisms may apply to multiple pathological conditions, as many diseases are characterized by both platelet and immune cell activation. For example, asthmatic patients showing elevated levels of platelet-adherent eosinophils exhibit higher serum levels of S1P compared to those patients with lower levels of platelet-adherent eosinophils, pointing towards a role of platelet-derived S1P in controlling eosinophil functions during airway inflammation [85].

Taken together, platelets link hemostatic and immune responses in several physiological and pathophysiological conditions [86], and S1P plays an important role as an autocrine and paracrine mediator linking platelet activation and inflammatory processes. Figure 4 summarizes the multiple paracrine effects of platelet-derived S1P and the pathophysiological responses associated with it. These may play a role in several clinical conditions, as discussed in the next section (Section 3). 

## 3. S1P and Platelets in Diseases

### 3.1. Diabetes Mellitus

Diabetes mellitus describes a group of metabolic diseases that are associated with hyperglycemia. The two main categories are type 1 and type 2 diabetes.

Type 1 diabetes (T1D) is characterized by an autoimmune destruction of insulin-producing beta-cells within the pancreas by CD4+ and CD8+ T cells and by islet-infiltrating macrophages [87]. Pancreatic beta-cells show an imbalance in the enzymatic capacity of S1P formation by SphK1 and SphK2 and degradation by SPL [88]. T1D often includes average or increased HDL cholesterol (HDL-C) values. In patients with T1D, the apoM/S1P complex is shifted towards light HDL particles, which are increased in T1D. While apoM/S1P in dense HDL particles inhibited TNF-α-induced vascular cellular adhesion molecule-1 (VCAM1) expression, light HDL particles had no effect. These effects may foster the development of the increased cardiovascular disease risk associated with T1D evoked through apoM/S1P in dense HDL particles [89]. Conversely, studies also showed that the activation of S1PR1 in the diabetic vascular endothelium prevents monocyte/endothelial interactions and therefore functions in an anti-inflammatory manner [75].

Type 2 diabetes (T2D) is characterized by a relative insulin deficiency caused by pancreatic beta-cell dysfunction and insulin resistance in target organs [90]. Recent studies indicate a possible protective role of the SphK/S1P signaling pathway in T2D, supported by the observation that SphK activation improves the hepatic insulin signaling in obesity and diabetes [91]. Interestingly, the metabolism of S1P appears to modify insulin signaling in peripheral tissue. In particular, an adaptive role of S1P has been proposed to counteract the development of insulin resistance in muscle, adipose tissue, and the liver [92]. S1P promotes proliferation and reduces the apoptosis of pancreatic beta-cells. Specifically, the upregulation of S1PR expression results in a hypoglycemic effect by increasing the number of beta-cells and insulin levels [93]. In addition, physiological concentrations of extracellular S1P inhibit the cytokine-induced apoptosis of beta-cells, again pointing towards a potentially protective function of S1P in T2D [94]. In agreement with these findings, diabetic mice treated with S1P exhibited significant reductions in glucose tolerance, insulin resistance, and the number of apoptotic beta-cells compared with the untreated group [95]. There are also studies suggesting a causal role of S1P metabolism and signaling in insulin resistance in the liver and in adipose tissue [92]. S1P levels in plasma from T2D patients are significantly higher than in plasma from healthy individuals [61]. This might be due to the association of diabetes with platelet hyperreactivity, which is reflected in a relative loss of efficacy of antiplatelet agents [96,97]. Increased platelet sensitivity and the associated increased release of platelet-derived S1P at the vessel wall may also predispose T2D patients to an increased risk of thrombosis and the development of vascular lesions, well-known complications of T2D.

As elaborated above, there is evidence for an important impact of S1P in type 2 diabetes. However, the direct involvement of platelet-derived S1P during diabetes-associated complications, although very likely, has to our knowledge been addressed experimentally only sparsely. Interestingly, the study from Russo et al. [98] indicated that platelets from diabetic patients released less S1P than healthy platelets when mechanically or chemically stimulated in vitro. In addition, the cardioprotective effects of platelets from healthy individuals depend on the platelet’s capacity to activate cardiac S1P receptors and the ERK/PI3K/PKC pathways. However, diabetic platelets release less S1P and lose their cardioprotective potential [98].

Metabolic syndrome (MetS) is a cluster of diseases and symptoms that can lead to diabetes. S1P may play multiple roles in the development of these diseases, mainly mediated by S1PR1 and S1PR3. Because S1P has both anti-inflammatory and pro-inflammatory effects and hypertensive versus hypotensive actions, its function remains controversial. In fact, S1P is linked to obesity [99]. Plasma S1P levels are positively correlated with body mass index (BMI), total body fat percentage, and waist circumference [100]. Studies have shown that S1P significantly decreases preadipocyte differentiation into adipocytes as well as the downregulation of adipogenic differentiation markers through multiple pathways [101]. Since inflammatory processes are also involved in MetS pathogenesis, S1P signaling pathways may contribute to the pathogenesis of metabolic diseases. S1P is associated with the upregulation of both pro-inflammatory and anti-inflammatory molecules. One possibility is that S1P increases the expression of pro-inflammatory factors in early stages. These reactions may lead to protective mechanisms and may help to mitigate the onset of obesity [102].

### 3.2. Vascular Lesions and Platelet-Derived S1P

Atherosclerosis is a chronic cardiovascular disease caused by lipid deposition at the vessel wall and is characterized by slowly progressing plaque formation and chronic inflammation of the arteries, which becomes clinically manifest when it triggers thrombosis [103]. Lipid deposition caused by endothelial injury, abnormal lipid metabolism, and hemodynamic changes in the susceptible sections of arteries develops into the formation of an atherosclerotic plaque [104]. In this process, ECs become activated and express inflammatory factors, attracting lymphocytes and monocytes to the endothelium. In turn, these adherent immune cells infiltrate the vessel wall, aggravating the inflammatory status of the vessels. Therefore, inflammation plays a central role in the atherosclerotic process [104].

Monocyte recruitment and adhesion to the vascular endothelium are key events in atherosclerosis. Platelets also adhere to ECs and contribute to the recruitment of leukocytes. The interaction between platelets and immune cells is triggered by the platelet production and secretion of inflammatory molecules, leading to activated immune cells being involved in the local vascular inflammation [105,106]. The activation of platelets results in increased local S1P concentrations. In this context, S1P affects lesion progression and thrombus formation in multiple ways. S1P acts synergistically with other factors and potentiates, for example, thrombin-induced tissue factor (TF) expression in ECs. Although it has not been proven directly, since S1P in solution, rather than platelets, was used experimentally, it is very likely that platelet-derived S1P leads to the propagation of thrombus formation at sites of EC injury [107]. However, as described earlier, pro- and anti-atherogenic effects of HDL-associated S1P have been reported. Hence, the influence of S1P on the development of atherosclerosis remains controversial. Depending on the S1PR subtype, the source of S1P (circulating within the plasma or locally released at sites of injury), the S1P concentration, and the affected target cell, S1P appears to exert both pro- and anti-atherogenic effects [108,109]. Recently, a preclinical study investigated the impact of fingolimod in rodent models of stroke, with age or atherosclerosis as comorbidities [110]. The authors found an improved post-ischemic outcome with fingolimod: fingolimod-treated hyperlipidemic mice showed a decreased infarct size but no difference in behavioral performance. They conclude that the effects of fingolimod in stroke are less robust than the existing literature might indicate and may depend on the inflammatory status of the animals [110].

In rabbits, a cholesterol-rich diet resulted in platelet hyperaggregability in response to low doses of agonists as well as in the development of hypercholesterolemic atherosclerosis. Under these conditions, the generation and release of S1P from platelets were significantly increased, pointing towards the involvement of platelet-derived S1P in the development and/or progression of cholesterol-stimulated lesion formation [111]. In addition, studies from Urtz et al. suggested that intrinsic S1P release through platelet Sphk2 controls platelet aggregation and thrombus growth in vivo. In a mouse arterial thrombosis model, they showed severely impaired thrombus stability in Sphk2 null mutants [45]. In contrast, other studies found no effect of the S1P analogue FTY720 on atherosclerosis in ApoE-deficient mice on a regular chow diet [112] or in only moderately hypercholesterolemic LDL-R-deficient mice [113]. Therefore, the effects of S1P in modulating vascular lesion formation may be pronounced and reveal themselves under conditions of elevated cholesterol levels. Further experimental settings in LDL-R-deficient mice demonstrated that FTY720 inhibits atherosclerosis by modulating lymphocyte and macrophage function. FTY720 at a high dose lowered the blood lymphocyte count and decreased the plasma concentrations of proinflammatory cytokines, pointing towards a general and rather unspecific immunosuppressive effect in these studies [114].

### 3.3. Multiple Sclerosis

MS is a currently untreatable degenerative neurological disorder associated with increased platelet activation and prothrombotic activity. MS is an inflammatory disease of the central nervous system (CNS) that leads to demyelination and neurodegeneration. The available data indicate the excessive intravascular activation of circulating platelets, implicating coagulation processes and inflammation in the pathophysiology of MS [115,116,117,118].

The proinflammatory activity of platelets is caused by various mechanisms such as the release of inflammatory molecules following platelet activation, as described above. As the lipid mediator S1P is released in bulk from activated platelets and also at a basal level from resting platelets [63,119], S1P is likely to contribute to such inflammatory mechanisms during MS. On the one hand, S1P signaling pathways regulate lymphocyte trafficking, which is a main event in MS. On the other hand, platelet-derived S1P, among others, might contribute to the cellular crosstalk mechanisms of activated platelets, leukocytes, and ECs. There are currently few data on the platelet-associated S1P effect on MS, and the subject needs to be evaluated in more detail.

Studies providing at least some information about platelet-associated S1P functions are experiments that were carried out as part of the studies of available MS therapies. One possible drug therapy for MS is fingolimod. Fingolimod was the first structural analog of sphingosine approved as a medication for the relapsing forms of multiple sclerosis [120,121]. Like Sph, it is phosphorylated by SphK2 to the bioactive form of the drug. Presumably, platelets are the major source of the active fingolimod (FTY720-P). In contrast to bulk S1P secretion, the release is independent of platelet activation [122]. After oral intake and conversion into its bioactive form, fingolimod acts as a modulator of S1PRs (mainly S1PR1), resulting in reduced infiltration and circulation of lymphocytes into the CNS. It has been observed that, after one month of treatment with fingolimod, the level of circulating platelets in both men and women significantly decreased [123]. In murine models of atherosclerosis, FTY720 inhibits atherosclerotic lesion formation and induces lymphopenia [124]. The observation that rats pretreated with FTY720 showed a dose-dependent increase in ADP-induced platelet aggregation compared to controls, however, is contradictory [53]. In humans, a double-blind placebo-controlled study evaluated the effects of fingolimod on platelet function in response to different stimuli ex vivo. Therefore, an increase in light transmission in platelet-rich plasma (PRP) in response to various concentrations of ADP, collagen, epinephrine, and ristocetin was utilized to measure the maximum platelet aggregation. Intriguingly, applied doses of 0.5 and 1.25 mg of fingolimod once daily over a period of one month in healthy volunteers did not affect the platelet function assessment in comparison to a placebo treatment [125]. While this does not point towards a critical impact of fingolimod on platelet functions in treated patients, the possible role of endogenous platelet-derived S1P in MS needs to be further deciphered.

There are also recent data linking the functions of the gut microbiome and alterations in mitochondrial function to MS. The permeability of the gut increases circulating lipopolysaccharides (LPS) through increasing toll-like receptor (TLR) activators. Therefore, nitric oxide synthase and superoxide are increased, which leads to peroxynitrite-driven increases in a-SMase and ceramide, which are associated with a decrease in the gut-microbiome-derived fatty acid butyrate. Butyrate is a histone deacetylase (HDAC) inhibitor that acts to regulate platelet and mitochondrial functions and suppresses the levels and actions of ceramides [126,127]. These circadian and gut-microbiome-derived changes are important to note, as they impact immune cell as well as platelet functions, with relevance across most medical conditions. As such, the nature of platelet function may be coordinated with variations in immune cells, including from circadian and gut-microbiome-derived products [128].

Alterations in the gut and circadian system do not only affect MS but may also affect the other conditions highlighted in this manuscript (diabetes and vascular lesions/plaques). Gut-microbiome-derived trimethylamine N-oxide (TMAO) has been classically linked to platelet hyperreactivity and is associated with an increased risk of atherogenesis and stroke [129], although not all data support this [130]. As TMAO increases ceramide and therefore may impact on the contrasting effects of ceramide and S1P at different S1P receptors, it will be important for future research to clarify these gut-microbiome-derived product effects in platelets and their concurrent effects on immune cells, including the consequences for the interactions of platelets and immune cells as well as the relevance of variations in the regulation of S1P efflux and effects.

## 4. Conclusions

The importance of platelets goes well beyond their role in stopping acute bleeding. Platelets flow through all organs and provide a link between the blood and the immune system. Thus, platelets support the essential communication functions of immune cells within the immune system. More precisely, platelets affect inflammatory processes on or in the vascular wall, which can occur as part of acute injuries or chronic vascular diseases. A key property of platelets is their ability to form and release inflammatory mediators. S1P is such an inflammatory mediator, or rather a modulator of inflammatory processes, since S1P can trigger both pro-inflammatory and anti-inflammatory responses. Platelet-derived S1P most certainly has a significant impact on direct vascular processes and diseases such as atherosclerosis in which the vasculature has a direct influence. As already discussed in detail, there are unfortunately few studies on the effect of platelet-derived S1P in various diseases. It can be assumed that the S1P released from platelets plays a decisive role in the pathogenesis of diseases with increased platelet reactivity and therefore elevated immune cell involvement. In fact, this needs to be examined more closely in future studies.

Therefore, at present, the immunological properties of blood platelets and their mediators are not the focus of therapeutic approaches. This should be improved or changed in the future. In addition to inhibiting the hemostatic function of platelets to prevent thrombosis, blocking the paracrine release of proinflammatory and proaggregatory mediators such as S1P could lead to an improved therapy for various diseases such as cardiovascular or autoimmune diseases. While selective S1PR agonists or antagonists may prove beneficial in this regard in the future, another possibility might be to reduce S1P release from platelets, for example, by the specific inhibition of MRP4. This may be possible without a fundamental or additional increase in the risk of bleeding, as this is the case to date with the use of conventional inhibitors of platelet function. The development and exploration of new substances that aim to change the immunological functions of blood platelets can be a new therapeutic approach in this regard.

## Figures and Tables

**Figure 1 ijms-23-10278-f001:**
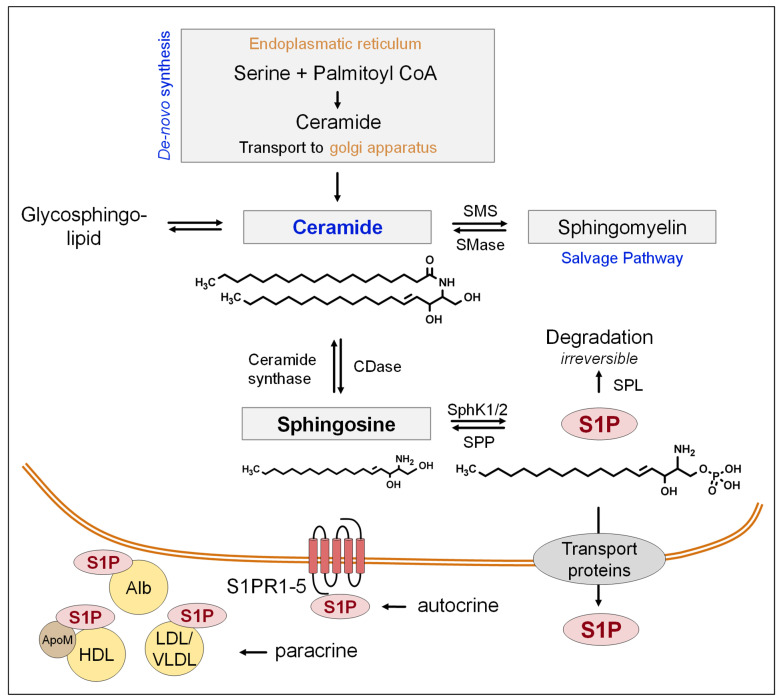
Metabolism of S1P, its cellular release, and binding partners. Sphingosine is formed via CDase from ceramides, which in turn can be formed de novo from serine and palmitoyl-CoA or sphingomyelin and glycosphingolipids. S1P is then formed intracellularly via SphK1/2 from sphingosine and transported out of the cell by ATP-binding cassette transporters (particularly ABCC4/MRP4) or the sphingolipid transporter 2 (Spns2) and the major facilitator superfamily transporter Mfsd2b. S1P can exert either paracrine or autocrine effects by binding to one of its G-protein-coupled receptors, S1PR1-5. Alb, albumin; apoM, apolipoprotein M; CDase, ceramidase; HDL, high-density lipoprotein; LDL, low-density lipoprotein; S1P, sphingosine-1-phosphate; SMase, sphingomyelinase; SMS, sphingomyelin synthase; SphK1/2, sphingosine kinase 1/2; SPL, S1P lyase; SPP, S1P phosphatase; S1PR1–5, S1P receptors 1–5; VLDL, very low-density lipoprotein.

**Figure 2 ijms-23-10278-f002:**
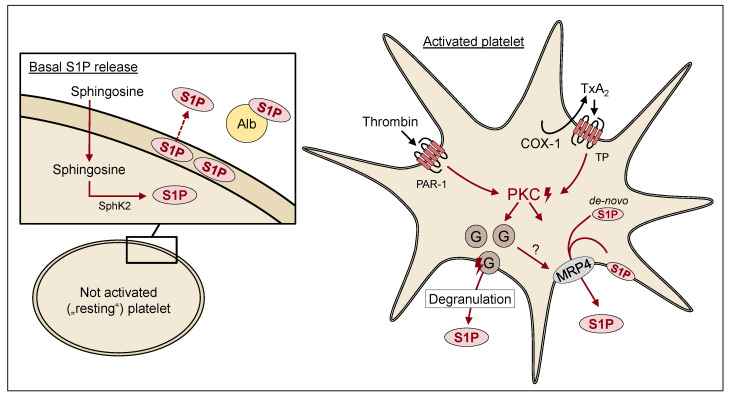
Release of S1P from resting and activated platelets. Platelets rapidly take up sphingosine from the serum and convert it into S1P. S1P is then stored within the platelet membrane and is released at a basal level in a calcium-dependent manner. After the activation of platelets with various agonists, such as thrombin or TxA_2_, S1P is released in bulk via de novo synthesis or from intracellular reservoirs in granules. The transporter MRP4, which is located mainly in the plasma membrane in activated platelets, has been shown to be capable of active S1P transmembrane transport. While both processes of degranulation and S1P secretion are linked to PKC, it is currently unclear how MRP4 is involved in the granule-dependent S1P release or if the secretion of S1P by MRP4 occurs independently. In addition, other mechanisms have been suggested to be involved in platelet S1P release, including the transporter Mfsd2b. For references, see Section 2.2. Alb, albumin; COX-1, cyclooxygenase 1; G, granules; MRP4, multidrug resistance protein 4; PAR-1, protease-activated receptor 1; PKC, protein kinase C; S1P, sphingosine-1-phosphate; SphK2, sphingosine kinase 2; TP, thromboxane receptor; TxA_2_, thromboxane A_2_; ?, unclear to date.

**Figure 3 ijms-23-10278-f003:**
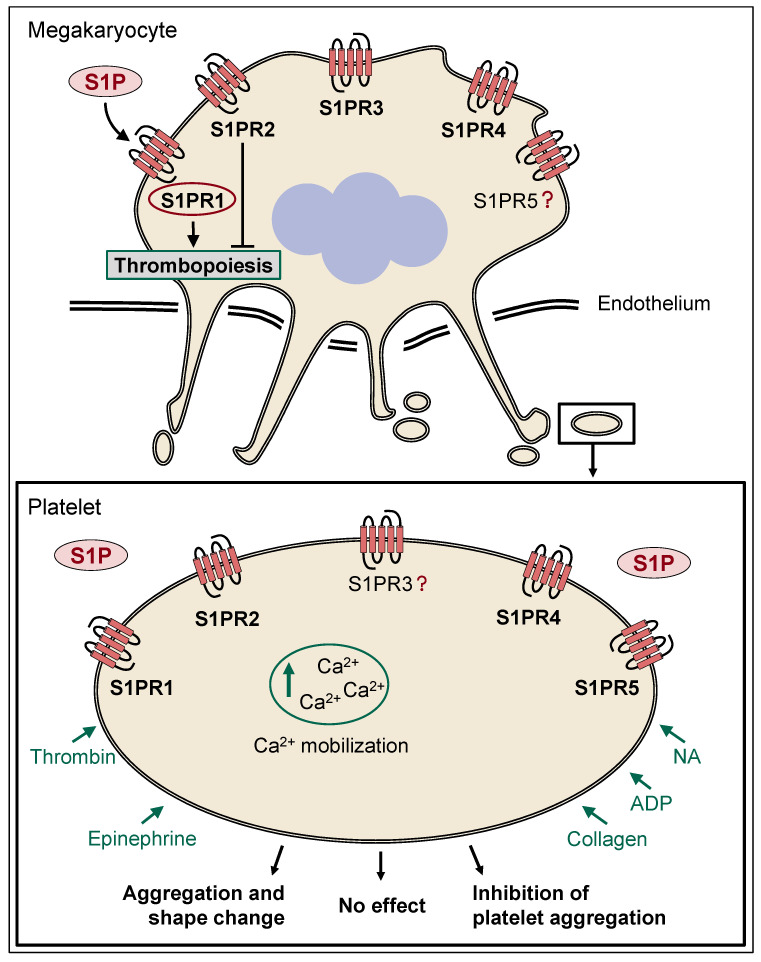
Effects of S1P on megakaryocytes and platelets. At present, the expression of S1PR1, 2, 3, and 4 on MKs has been detected. S1P regulates thrombopoiesis through S1PR1. However, thrombopoiesis is suppressed by the activation of S1PR2. Different autocrine effects on platelet activation are discussed. Evidence suggests the possible involvement of S1PR1, 2, 4, and 5 in platelet responses, indicating that presumably all members of the S1PR family are present in MKs/platelets. In addition to no effect or even inhibitory effects on platelet activation, a platelet-activating or aggregation-promoting effect of S1P on platelets when combined with thrombin, epinephrine, NA, ADP, or collagen has been described. In addition, an increase in cellular Ca^2+^ levels after S1P stimulation is under discussion. ADP, adenosine diphosphate; MK, megakaryocytes; NA, noradrenalin; Ca^2+^, calcium; S1P, sphingosine-1-phosphate; S1PR1–5, S1P receptors 1–5; ?, unclear to date.

**Figure 4 ijms-23-10278-f004:**
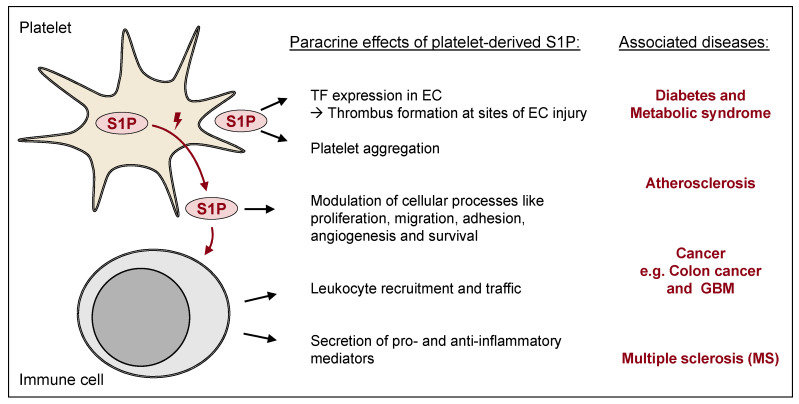
Paracrine effects of S1P on endothelial and immune cells. S1P released from platelets (in bulk during platelet activation 🗲) can affect various cells and cell functions. The focus in this scheme is on the S1P effects on EC or platelets as well as on immune cells, which are attracted by S1P and then release pro- and anti-inflammatory mediators. S1P thus has a direct or indirect influence on various diseases, such as diabetes, atherosclerosis, and MS. EC, endothelial cell; GBM, glioblastoma; MS, multiple sclerosis; S1P, sphingosine-1-phosphate; TF, tissue factor.

**Table 1 ijms-23-10278-t001:** Literature overview of the effects of S1P on platelets or megakaryocytes and the possible S1PR involved. ADP, adenosine diphosphate; AP, activating peptide; Ca^2+^, calcium; MK, megakaryocyte; NA, noradrenalin; PAR-4, protease-activated receptor 4; PPP, platelet-poor plasma; PRP, platelet-rich plasma, S1P, sphingosine-1-phosphate; S1PR, S1P receptor; TRAP, thrombin-receptor-activating peptide.

Involved Receptor	Cell Type	Findings
S1PR1, 2, and 3	MK	Expression of S1PR1,2, and 3 mRNA in mature MKs was detected. S1P regulated thrombopoiesis through S1PR1 in mice [52].
S1PR1 and 2	MK	In mice, S1PR1 can suppress megakaryopoiesis and, additionally, S1PR2 signaling can suppress platelet production [53].
S1PR4	MK	S1PR4 was upregulated during the development of human MKs from progenitor cells [66].
Unclear to date	Platelets (PRP)	S1P alone did not stimulate aggregation but inhibited the TRAP- and NA-induced aggregation [64].
S1PR4	Platelets (PRP)	Collagen-stimulated platelet aggregation was suppressed by CYM50260 (selective S1PR4 agonist) [65].
S1PR2	Platelets (washed)	S1PR2 expression was detected via Western blotting. S1P leads to an increase in intracellular Ca^2+^ and platelet aggregation, presumably due to S1PR2 activation [61].
Unclear to date	Platelets (washed)	S1P alone did not affect platelet aggregation [63].
Unclear to date	Platelets (washed)	S1P induced platelet aggregation and shape change. Weak agonists such as epinephrine and ADP synergistically induced platelet aggregation, possibly due to intracellular Ca^2+^ mobilization [25].
S1PR1	Platelets (whole blood)	Platelet aggregation was triggered through the S1PR1 agonist SEW2781. Platelet aggregation triggered via S1PR1 might be amplified through PAR-4 and ADP [45].
S1PR1, 4, and 5	Platelets (washed)	S1P alone did not affect platelet aggregation. In a concentration-dependent manner, S1P had a biphasic effect on PAR-1-mediated platelet aggregation. PAR-1-mediated platelet aggregation was significantly increased through the S1PR1 agonist SEW2871. S1PR4 (CYM50260) and S1PR5 (A971432) agonists caused the inhibition of high-PAR-1-AP-mediated platelet aggregation [64].

## Data Availability

Not applicable.

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
