# Peer review of "Platelet-Derived S1P and Its Relevance for the Communication with Immune Cells in Multiple Human Diseases"

_ijms, 2022, doi:10.3390/ijms231810278_

Round 1

Reviewer 1 Report

Review articles on platelet sphingosine-1-phosphate are already available. In my opinion the submitted paper does not bring much new to the already available summaries. The main subject given in the title of the article and the main idea of ​​presenting the role of platelet-derived S1P in the regulation of communication of blood platelets with immune cells in various diseases is not reflected in the content of the article and it can be found only in lines 301-314. The rest of this article presents loosely coupled descriptions of what S1P can do in the immune system (and beyond), and how S1P is metabolized in blood platelets, but the real link between immune system and blood platelets through S1P is not really presented.

In most of the paragraphs we find only brief references to platelets, and these are mainly unproven hypotheses.

Too often Authors make exaggerated hypothetical assumptions. For example, citing Takeya et al. (2003) in which, as far as I know, no platelets were used at all, but only S1P solution, the Authors suggest that it is platelet S1P that increases TF expression. This is of course very likely, but still not experimentally proven.

I think the main problem is that the Authors have undertaken to write a review on a subject that has a weak experimental foundation and still little evidence.

Reviewer 2 Report

In this manuscript by Tolksdorf and colleagues, a comprehensive overview of the role of platelet-derived S1P in human disease is provided. The manuscript is clear and well written, and covers the diverse aspects of S1P biology very well. A number of comments are listed below for further improvement of the manuscript:

In figure 1: some structure formulas (Sph, S1P and ceramide) would further illustrate biogenesis. 

Which metabolic route is involved in the de novo biosynthesis of S1P during platelet activation?

What is the tissue expression pattern of the ABCC4 transporter?

S1P appears to modulate sensitivity of other GPCR ligand-related responses (e.g. CXCR4, PMID: 17158356). Could there be a formation of heterodimers between the S1PR types and other GPCR?

Line 173: The exact / precise role of S1P (makes sentences connect better) 

Line 179: here fingolimod is already introduced and in the section about MS, it is explained more elaborately. I suggest moving that section to line 179.

A listing of the S1P receptors expressed on platelets (optimally along with the EC50 values for S1P and FTY720P) could be useful to better understand the section on autocrine effects of platelets (starting from line 181).

There are some interesting studies on S1P in transfusion-related lung injury: e.g. PMID: 33232973 worth discussing.

Line 412-414: Platelet function testing is deceptively complicated. Perhaps it is worth a brief mention (e.g. PMID: 32522065).

Conclusion section: how do the authors envision a manipulation of platelet S1P levels (or release potential)? This would be an interesting approach to target a number of diseases.

Reviewer 3 Report

Tolksdorf et al. have provided an overview of the Platelet-derived S1P and its role in human health and disease. The review is concise, within the scope of the journal, and well-referenced. Illustrations are appropriate. I would like to congratulate the authors for providing this interesting review.

Some minor comments:
1) What is the role of S1P in the development of complications in subjects with T2DM? Is there any evidence linking this molecule with an increased risk of renal disease or coronary artery disease in DM, for example?

2) Please elaborate more on human studies of S1P in the mentioned pathologic states. Could this end up being a biomarker of their presence/complications/prognosis? It would also be interesting to see a table summarizing the evidence of the studies mentioned in the text.

3) Have preclinical studies assessed the therapeutic modulation of S1P in the chosen diseases? If so, it could be added in the corresponding sections.

Round 2

Reviewer 1 Report

I appreciate the amount of work the Authors have done. But I still believe that despite the many improvements made to the second version of the manscript, the article is still loaded with loosely related facts or too far-reaching hypotheses. I realize that a review article can be an appropriate place for such speculation or for the delineation of future experiments, but there should be a balance between already established facts and hypotheses. Furthermore, I still maintain my opinion that the main topic of the article (the role of platelet-derived S1P in regulating immune cell responses) is only a fraction of the content of the article.

Author Response

Reply: We thank the reviewer for his critical opinion and that he appreciates the amount of work and adjustments we have perform on the revised manuscript. In the revised manuscript version, we tried to further connect the individual aspects of the work and to smoothen the transitions between the content topics. This is not an easy task, since that we have also taken into account the comments of the other reviewers and of the academic editor. We hope that the revised version of the manuscript now better meets the opinion of the reviewer.